# Analysis of rural broadband adoption dynamics: A theory-driven agent-based model

**Ankit Agarwal, Casey Canfield** *

Department of Engineering Management and Systems Engineering, Missouri University of Science and Technology, Rolla, Missouri, United States of America

\* canfieldci@mst.edu

## Abstract

Demand for broadband internet has far outpaced its availability. In addition, the "new normal" imposed by the COVID-19 pandemic has further disadvantaged unserved and underserved areas. To address this challenge, federal and state agencies are funding internet service providers (ISPs) to deploy broadband infrastructure in these areas. To support goals to provide broadband service to as many people as possible as quickly as possible, policymakers and ISPs may benefit from better tools to predict take rates and formulate effective strategies to increase the adoption of high-speed internet. However, there is typically insufficient data available to understand consumer attitudes. We propose using an agent-based model grounded in the Theory of Planned Behavior, a behavioral theory that explains the consumer's decision-making process. The model simulates residential broadband adoption by capturing the effect of market competition, broadband service attributes, and consumer characteristics. We demonstrate the effectiveness of this type of tool via a use case in Missouri to show how simulation results can inform predictions of broadband adoption. In the model, broadband take rates increase as the presence of existing internet users in the area increases and price decreases. With further development, this type of simulation can guide decision-making for infrastructure and digital literacy investment based on demand as well as support the design of market subsidies that aim to reduce the digital divide.

## 1. Introduction

Addressing the issue of inequitable access to high-speed internet, commonly referred to as the "digital divide," demands substantial and sustained government investment [1]. Government agencies and internet service providers (ISPs) grappling with this challenge encounter two primary hurdles in bridging the gap. First, many underserved and unserved communities are situated in rural areas characterized by lower population densities and reduced median household incomes. The deployment of broadband services in such regions increases the cost per household, reducing affordability [2]. This poses unattractive revenue prospects for larger for-profit ISPs, and the substantial initial infrastructure costs further compound the challenges in bringing broadband to rural areas, even for non-profit ISPs. Second, areas lacking adequate access often exhibit lower levels of digital literacy due to a higher representation of low-income

**Data Availability Statement:** Data and code can be found in a GitHub repository, https://github.com/ankeeet-agarwal/Broadband-ABM.

**Funding:** The author(s) received no specific funding for this work.

**Competing interests:** The authors have declared that no competing interests exist.

households and lower rates of formal education [3]. As of 2021, statistics indicate that only 72% of rural households own personal desktops/laptops, in contrast to the 80% ownership rate observed in urban households [4].

The successful deployment of broadband services in unserved and underserved areas, with the aid of federal financial assistance, hinges on consumer adoption. Achieving high adoption rates can be particularly challenging, given that rural communities vary significantly in terms of their unique needs, digital readiness, population density, geography, and other factors [5]. Consequently, catering to the diverse requirements of rural households requires a tailored approach, as a one-size-fits-all solution falls short. For instance, in 2018, fixed wireless and cable internet providers gained an advantage over fiber-based services due to the evaluation criteria employed by the Federal Communications Commission (FCC) in the Connect America Funds (CAF-II) bidding process [6]. Typically, wireless services face technological limitations in terms of last-mile bandwidth availability, line-of-sight requirements, and limited potential for future upgrades compared to fiber technology. This makes them less suitable for communities where barriers like uneven terrains, dense tree cover, and high end-user bandwidth demands exist [7]. Nevertheless, due to the higher installation cost per mile associated with fiber internet, lower-tier speeds (ranging from 25 to 100 Mbps) provided by fixed wireless ISPs were given preference. Consequently, the ISPs that received the majority of the funds delivered lower speeds at a higher cost to consumers [8]. This outcome likely undercuts the intended impact of federal financial assistance, perpetuating access inequities.

In response to the COVID-19 pandemic, there are many financing programs initiated by state and federal governments to fund the fixed costs of deploying high-speed internet access in unserved areas. For example, the Infrastructure Investment and Jobs Act allocated $65 billion, the American Rescue Plan allocated $20.4 billion, and the Consolidated Appropriations Act allocated $1.6 billion to deploy infrastructure, subsidize monthly bills, and provide digital literacy training [9]. The ISPs' willingness to provide broadband access needs to be coupled with a comprehensive understanding of community's needs to maximize the intended impact of federal assistance. For example, this requires understanding which technology is most suitable for improving service availability in the area, the current status of internet use in the community, and which attributes of the broadband service need attention to achieve maximum adoption [10, 11]. Despite the diversity of needs, engaging with each community through surveys or discussions would be inefficient [12, 13]. This creates an opportunity to use simulation-based approaches capable of capturing the heterogenous needs and complexities of adoption decisions, to increase the efficiency of that spending and determine appropriate counterfactuals for evaluation.

This research aims to address two primary research questions:

1. How can a theory-based simulation capture broadband adoption dynamics in served and unserved communities?

2. How do attributes of the broadband technology and population influence adoption dynamics?

The primary contribution of this work is in demonstrating an approach for scaling modeling efforts for broadband adoption in data poor contexts. In the broadband context specifically, this requires modeling market competition, social network effects, digital literacy, and affordability [14].

In this research, we employ an agent-based model (ABM) simulation to forecast the uptake of broadband services within the context of market competition and consumer attributes. ABM constitutes a simulation methodology in which agents, each characterized by unique

traits, engage with one another and their surroundings, culminating in emergent outcomes [15]. These agents possess diverse motivations, preferences, and attributes that steer their actions. From these micro-level behaviors, larger-scale system-level consequences can manifest. This bottom-up approach affords ABM a distinct advantage over top-down modeling techniques, such as equation-based models, particularly in representing intricate non-linear systems influenced by human behavior [16]. ABM has found extensive applications in diverse technology adoption scenarios [17]. These encompass a wide spectrum, ranging from the adoption of rooftop photovoltaics [18, 19] to the implementation of smart metering systems [20] and even community responses to mining activities [21]. ABMs featured in the technology adoption literature typically draw upon established behavioral theories and the principles of utility maximization to formulate the rules and underlying assumptions governing agent decision-making processes [22].

This ABM draws upon the Theory of Planned Behavior (TPB) to forecast consumer interest in subscribing to high-speed internet services. Within this model, we take into account factors such as affordability, digital literacy, consumer attitudes towards various attributes of the service, and the impact of social influence. Furthermore, we conduct a sensitivity analysis to gauge the degree of influence exerted by each input parameter. Our findings underscore the suitability of theoretically grounded models in the context of modeling broadband adoption within rural markets, where acquiring market-specific data can prove to be a formidable challenge. In a broader context, our model displays a pronounced sensitivity to two key factors: the proportion of active internet users in a given area and the pricing structure of the newly introduced service. This suggests that a higher prevalence of internet users in a region and a reduction in the monthly cost, potentially through subsidies, have the potential to significantly bolster adoption rates. The subsequent sections of this document provide a review of the broadband adoption literature (Section 1.1), describe the present model (Section 2), highlight key findings (Section 3), compare our findings to the literature (Section 4), and identify the policy implications of this research (Section 5).

## 1.1. Modeling broadband adoption

Broadband adoption hinges on a multitude of factors, encompassing both individual household attributes and broader environmental considerations. Household-level factors include elements such as the perceived advantages of internet access, opinions regarding local Internet Service Providers (ISPs), digital literacy levels, and affordability. In Canada, rural households perceived broadband as less beneficial due to their limited access. However, when access levels were equalized, rural households began to place a higher value on e-services compared to their urban counterparts [23]. In Jordan, internet users exhibit greater loyalty to their chosen ISP when they find satisfaction in service performance and pricing [24]. An investigation targeting non-adopters suggested that households with computer access, those situated in remote locations, or those belonging to minority communities were more likely to embrace broadband adoption if the service was reasonably priced. This study projected a potential 10% surge in demand if the price was reduced by 15% [25].

Considering the increasing bandwidth requirements of web applications over time and the inadequate availability of internet services in rural areas, it is important to investigate how consumers' perceptions of the existing options in their areas have evolved [26]. Particularly during the Covid-19 pandemic, when remote work and online education became prevalent, rural consumers were disproportionately affected compared to their urban counterparts [27]. This temporal aspect poses a limitation to survey-based approaches, as they can only capture consumers' preferences at a specific point in time. This limitation can be overcome by

employing simulation-based methods, which can adapt to changing market dynamics and consumer preferences. By focusing on digital literacy and the affordability of services, policymakers can address the specific needs of rural areas to ensure equitable access to broadband services.

Broadband adoption is sensitive to external environmental factors, such as the cost and quality-of-service, of the network. A U.S. study found that willingness-to-pay increased as speed and reliability increased [28]. However, there is also evidence that demand for broadband is inelastic over time. Analysis of market data suggests that small increases in speed are not associated with higher willingness to pay [29]. These studies suggest that there is a complex relationship between broadband access and adoption, which may be sensitive to threshold or tipping point effects. Given the complex dynamics of consumers' adoption decisions, ABM would be a more appropriate choice than equation-based modeling (EBM) approaches. EBM focuses on capturing the overall behavior of the system through mathematical equations. It often represents the system at an aggregate level, using average values or statistical distributions. The representation is highly simplified by assuming homogeneous behavior across the system. Therefore, it may not explicitly capture emergent phenomena arising from individual interactions. ABM allows for a high level of detail and heterogeneity by representing consumers as individual agents. Each agent can have unique attributes, preferences, and decision-making rules. This enables ABM to capture the diversity of consumer characteristics and behaviors, which is crucial in modeling adoption decisions influenced by individual choices. EBM is more rigid and less adaptable since it relies on predefined equations. Changes or additions to the model structure or behaviors may require substantial modifications to the equations. ABM offers greater flexibility and adaptability compared to EBM [15]. Consumer behavior and decision-making processes can be highly context-dependent and subject to change. ABM allows for easy incorporation of new rules, behaviors, and factors that influence adoption decisions. This flexibility enables researchers to explore various scenarios and test the effects of different variables on adoption outcomes. In the context of broadband adoption, other studies have combined ABM with a systems dynamics model [30] and wireless signal model [31–33] to improve accuracy by using ABM to model the human and stochastic parts of the system.

Various behavioral theories have been employed to investigate the factors influencing broadband adoption. These include well-established models such as the Technology Acceptance Model, Diffusion of Innovation theory, and the TPB. For instance, in Korea, the decision to adopt broadband was influenced by factors such as compatibility with individual needs, the trialability of the technology, and the visibility of its usage [34]. In India, broadband adoption was shaped by considerations related to social outcomes, service quality, the availability of supportive resources, and the ability to utilize Internet applications effectively [35]. In the United Kingdom, social influence, perceived resource availability, and self-efficacy were key determinants of broadband adoption [36]. Across different countries, the constructs defined by the TPB emerge as the most commonly referenced factors. As a result, we have chosen to focus our current model exclusively on TPB. Additionally, to enhance our model's representation of social influence and outcomes, we have incorporated a small-world network.

**1.1.1. Theory of planned behavior.** According to TPB, attitude, subjective norms, and perceived behavioral control predict behavior [37]. Attitude relates to an individual's personal assessment of the behavior at hand. Subjective norms are the influence exerted on an individual by their surroundings, encompassing factors like peer effects and information received via large-scale communication channels. Perceived behavioral control pertains to an individual's ability to perform the behavior in question. Consequently, if an individual harbors a favorable opinion of a given behavior, observes their peers actively participating in it, and possesses the

means to carry it out, they are more likely to harbor the intention to engage in the behavior themselves. The applicability of TPB extends across a diverse spectrum of contexts, including predicting behaviors such as smoking cessation [38], dietary choices [39], and the adoption of Internet banking [40].

In the realm of technology adoption models within ABM, TPB has gained prominence and has presented itself as an improvement over earlier Diffusion of Innovation (DoI) models. DoI models presupposed rational consumers with uniform preferences [41], emphasizing the characteristics of adopters (e.g., innovators, early adopters) while assuming uniform behavior within each category. This oversimplification neglects the nuanced interplay of individual-level disparities in beliefs, preferences, and decision-making processes. TPB, in contrast, underscores the significance of individual decision-making processes by acknowledging the variability in the influence of psychological constructs on adoption decisions. ABM, with its capacity to model individual agents and their decision rules, aligns with the TPB's focus on how individual beliefs and intentions propel behavior. This alignment enables a more detailed and realistic portrayal of consumer decision-making processes in ABM. Furthermore, DoI primarily assumes an individual's capacity to engage in a behavior is shaped by the influence of their social networks. In broadband adoption scenarios, factors such as affordability and digital readiness play pivotal roles in decision-making. TPB incorporates the concept of perceived behavioral control, mirroring an individual's belief in their capacity to execute a behavior. TPB-based ABM can explicitly incorporate this concept by integrating agent-level attributes like self-efficacy or the perceived level of control over the adoption decision. In TPB-based ABM, agents appraise the utility of a technology through the lens of their distinctive preferences, reflecting the attitude construct of TPB [42]. Subjective norms manifest as social utility, representing the perceived utility of an agent's neighbors within their social network. Perceived behavioral control often functions as a constraint encompassing an agent's financial capacity and/or operational ability concerning the technology [42]. In certain instances, agents' intrinsic attributes can be initialized through survey data, further enhancing the model's fidelity [43].

**1.1.2. Social network.** Prominent network theory graphs find application in delineating agent-agent interactions and the rules governing information diffusion, aiding in the characterization of the presence, frequency, and strength of interactions [16]. Random graphs imbued with small-world attributes have shown the potential to optimize clustering coefficients and average path lengths [44, 45]. These small-world networks can encompass any lattice structure with n nodes and k edges per node. Certain edges are randomly rewired based on a rewiring probability denoted as 'p'. Small-world networks are characterized by short average path lengths, facilitating efficient communication, information dissemination, and navigation, as fewer steps are needed to reach any other node. Conversely, random graphs typically exhibit longer average path lengths, necessitating more steps to reach distant nodes, which results in less efficient global connectivity. Small-world networks also feature a high clustering coefficient, indicative of tightly connected clusters or communities. In contrast, random graphs have a low clustering coefficient, signifying a lack of local clustering and sparser node connections, which diminishes local cohesion and specialization. For representing rural social networks, it is most suitable to employ small-world networks with a degree of randomness, as this configuration introduces adequate local clustering, appropriate path lengths, and offers an improved depiction of spatially dispersed rural communities.

Small-world networks have found utility in ABMs for technology adoption, as they closely mimic the topological characteristics of real-world social networks [17]. In scenarios pertaining to renewable energy adoption, agent clustering is achieved through the measurement of similarity and proximity [18, 19]. This approach has also been applied in ABMs for simulating

the diffusion of telecommunications technology by adjusting the rewiring probability and the proportion of initial innovators [46].

## 2. Model overview

The current model makes predictions regarding broadband adoption by considering consumer preferences influenced by attitudes, social influence, and accessibility. Within the simulation framework, we delineate the interactions among agents and their environment. Over the course of the simulation, information about a novel broadband service spreads throughout a community. Households decide whether the revised perceived utility of the broadband service justifies its adoption, based on their status as existing users or non-adopters. The simulation unfolds over 12 time steps, with each step representing one month, thus encompassing a full year. The model is implemented using the NetLogo platform (Version 6.1.1) [47]. Data and code can be found in a GitHub repository [48].

### 2.1. Environment and agent-environment interactions

The environment within the simulation characterizes the available broadband service. To align with U.S. Census data, the simulation area is defined as a specific zip code. A broadband service is characterized by four key attributes: (1) speed, (2) monthly cost, (3) the presence of a data cap, which imposes limitations on the total data volume that can be transmitted in a month before throttling occurs, and (4) reliability, which is closely tied to the quality of customer service (see Table 1). The model is designed to represent a spectrum of communities, including unserved, underserved, or fully served, with or without competition, contingent upon the attributes of the existing and new broadband services. In an unserved scenario, there is either no broadband service or it is inadequately provided ($\leq 10$ Mbps). Conversely, an underserved scenario is characterized by the presence of an existing service that falls short of meeting the minimum high-speed definition ($\leq 25$ Mbps).

### 2.2. Agents and agent-agent interactions

The community's characteristics are determined by two primary factors: the total population size, which is related to population density, and the percentage of existing residential internet users. These values are derived from U.S. Census data, drawing upon information from the ACS 2018 Computer and Internet Use data to estimate the percentage of current users. We use TPB to describe individual agent decision-making processes. Each agent within the model represents a household, defined by its annual income, education level, and preferences regarding internet speed, monthly cost, data cap, and service reliability (as detailed in Table 2). The location of each household is randomly assigned.

**Table 1.** Summary of environment attributes.

| Environment Attribute | Notation | Value |
|---|---|---|
| Existing broadband service | 0 | Denoted by subscript, represents an unserved area if not assigned |
| New broadband service | 1 | Denoted by subscript, represents competition if there is existing service |
| Speed | $S$ | From ISP website [Mbps], normalized on 0–1 scale |
| Monthly cost | $C$ | Value based on service [$/month], normalized on 0–1 scale |
| Data cap | $D$ | Value based on service [GB/month], normalized on 0–1 scale |
| Reliability score | $R$ | From customer ratings in Google, Facebook, and BroadbandNow, represented as scale from 1–5 |

**Table 2. Summary of agent attributes.**

| Agent Attribute | Notation | Value |
|---|---|---|
| Household | $x$ | Denoted by subscript |
| Other household | y | Denoted by subscript |
| Total population size | $N$ | Model input [49] |
| Percent of existing users | $P$ | Model input [50, 51] |
| Annual household income | $inc_x$ | 14 levels [49] |
| Income factor | $if_x$ | Income level normalized on 0–1 scale |
| Highest household education level | $edu_x$ | 7 levels [49] |
| Digital Literacy factor | $ef_x$ | Education level normalized on 0–1 scale |
| Importance of speed | $s_x$ | Uniform (0,1), importance sums to 1 |
| Importance of monthly cost | $c_x$ | Uniform (0,1), importance sums to 1 |
| Importance of data cap | $d_x$ | Uniform (0,1), importance sums to 1 |
| Importance of reliability | $r_x$ | Uniform (0,1), importance sums to 1 |
| Existing user? | $p_x$ | Boolean, randomly assigned at initialization based on percent of existing users ($P$) |
| Information received? | $f_x$ | Boolean, represents diffusion of information |
| Perceived utility of new broadband service | $U_x$ | Calculated at each time step, see Eqs 2 and 3 |
| Social influence | $I_x$ | Calculated at each time step, see Eq 4 |
| Importance of utility of broadband service | $u_x$ | Uniform (0,1) |
| Importance of social influence | $i_x$ | Uniform (0,1) |

The demographic parameters for a specific geographic area are determined based on U.S. Census data, with the Census data specifying a distribution from which household agents are randomly assigned values. We assume a correlation between affordability and income, as well as between education and digital literacy. This alignment, in accordance with [18], is achieved by introducing an error term that interacts with the normalized demographic term. Wealthier and more educated households are assumed to be more likely to adopt broadband. Furthermore, each household has different preferences for internet speed, monthly costs, data caps, and reliability. For instance, some households might be willing to pay more if they receive their desired speed, while others might accept data caps if the costs are low. To account for these variations, each preference threshold is randomly assigned to each household, encompassing all potential combinations. Throughout the simulation, households are presented with multiple opportunities to determine whether to adopt broadband service. At the commencement of the simulation, a portion of users is randomly designated as existing users. Subsequently, households exhibit varying degrees of reliance on high-speed internet, quantified by their perceived utility of broadband service. Additionally, households respond diversely to social influence, ranging from resistance to receptivity.

Social influence, which encompasses agent-agent interaction, is established through the utilization of a small-world network [18, 19]. As illustrated in Fig 1, every household is subject to influence from neighboring households with similar demographic characteristics, specifically income and education. Additionally, random links play a role in the influence dynamics. In this context, "neighboring" is defined within a radius of 5 units, facilitating the formation of clusters within the small-world network. This clustering aspect replicates a significant topological feature observed in real-world social networks. The similarity between households is

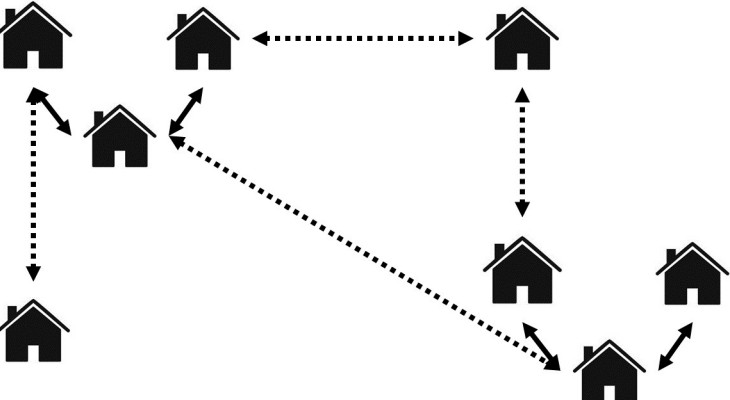

**Fig 1. Small-world network of agent households.** Each arrow represents a link that allows for social influence from similar nearby (solid lines) and random far away (dotted lines) households.

defined as:

$$\text{Similarity}_{ij} = \left[ 0.5 - \left| \left( \frac{\text{inc}_x - \text{inc}_y}{14} \right) \right| \right] + \left[ 0.5 - \left| \left( \frac{\text{edu}_x - \text{edu}_y}{7} \right) \right| \right] \quad (1)$$

where households form links with neighbors if the similarity is greater than 0.8. Prior research has typically employed a 95% similarity value [19]. Nevertheless, in this study, a lower value is adopted to promote a greater number of local connections, which is more in line with expectations in a rural community. Furthermore, half of the households establish connections with another randomly selected agent. These random connections account for other types of relationships, such as those with colleagues, classmates, and relatives, who may not necessarily reside in close proximity [18, 42].

## 2.3. Simulation

As outlined in Fig 2, the model is initialized at t = 0, drawing from the model inputs to determine the environment, agent households, and connections among agents. During this initialization phase, information pertaining to the new broadband service is randomly disseminated to 5% of the agents. It is assumed that these households have been exposed to advertisements regarding the new broadband service and have chosen to share this information with their peer households, i.e., those they are linked to. Consequently, as households receive this information, the parameter denoted as "Information received?" transitions to TRUE. At each subsequent time step, every household proceeds to revise its perceived utility of the broadband service and assess the influence of social interactions, culminating in a decision regarding the adoption of the new broadband service. The utility for existing users reflects how much the new service improves their utility above the existing service:

$$U_x = s_x(S_1 - S_0) - c_x(C_1 - C_0) + d_x(D_1 - D_0) + r_x(R_1 - R_0) \quad (2)$$

where the weights for each feature (speed, cost, data cap, and reliability) of the service sum to 1 ($b_x + c_x + d_x + r_x = 1$) to represent how important each feature is to that household. If $U_x \leq 0$, the household perceives the new broadband service as less favorable, but if $U_x > 0$, the household perceives the new service as superior. The weight for price has a negative sign so that the

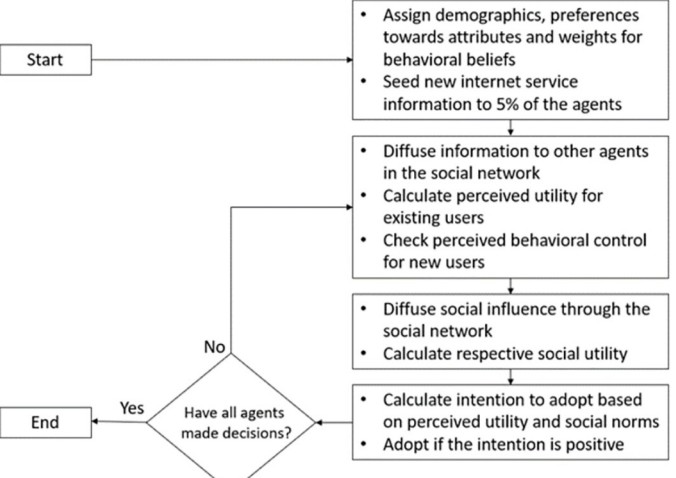

**Fig 2. Flowchart of the broadband adoption model.**

agent gets positive utility from the new service only when the price is lower than the existing service. If current service is cheaper, the agent's overall utility is reduced.

If the household is not an existing user, they must first establish perceived behavioral control, based on TPB. A household's means to execute adoption behavior, which are calculated as the summation of the affordability and digital literacy measures, has a threshold of 0.5. Below this threshold, the household does not participate in the newly introduced service [52]. This is not an issue for existing users who already own a similar product [53]. The utility for non-adopters measures the attitude that a household has towards the broadband service [53]:

$$U_x = \begin{cases} -1, & if_x + ef_x \le 0.5 \\ s_x S_x - c_x C_x + d_x D_x + r_x R_x, & if_x + ef_x > 0.5 \end{cases} \tag{3}$$

a household's favorable opinion of the new service is indicated by a higher value of $U_x$. If there is insufficient perceived behavioral control, the household has negative utility. Such agents would have positive intention to adopt only if the influence from other agents can offset the utility. The social influence of a household's network (based on all agent-agent links for a particular household) is determined by the average of how their peers perceive the utility of the new broadband service [54]:

$$I_x = \frac{\sum_{y=1}^{N} U_y}{N} \tag{4}$$

Higher number of peers with positive opinion of the new service increases the value of $I_x$, therefore making it more likely for a household to adopt the new broadband service. Ultimately, the intention to adopt the new broadband service depends upon the perceived utility, social influence, and the associated weights [20, 42]:

$$Intention_x = u_x U_x + i_x I_x \tag{5}$$

If $Intention_x > 0$, the agent is considered to have adopted the new service. Each simulation run is evaluated in terms of the count of new service subscribers and the percentage of new service subscribers. To construct a confidence interval, the simulation is repeated 10,000 times.

## 3. Results

We demonstrate using an ABM to predict the number and percent of subscribers for (1) a served and (2) an unserved community with competition between providers. In addition, a sensitivity analysis shows how input parameters influence the model outcomes.

### 3.1. Predicting take rate in a served community

In Northeast Missouri, Perry is a small community of 274 households. The median household income is $38,000 and the average resident has completed a high school degree. Approximately, 14% of the population is below the poverty line. Ralls Technology Fiber Solutions, which is a subsidiary of a rural electric co-operative, launched a new fiber service in Perry in November 2019. At that point in time, 70% of the households had existing access to a DSL service provided by CenturyLink [51]. The community characteristics are summarized in Table 3. The attributes of the services in Table 4 match those of Ralls Technology and Century-Link at the time.

In the simulation, after 1 year of access to both the fiber and DSL services, 68% of households had adopted fiber service (Fig 3A). There is a 14% increase in household-level internet adoption as 40 additional households adopted the new service. The DSL subscribers reduced from 191 to 35, which is from 70% to 12% of the population (Fig 3B). The largest drop in DSL subscribers is observed in the first 4 months as there is also a steep increase in fiber subscribers. Fig 3B shows higher uncertainty (as indicated by the shading) in predicting the adoption/switching behavior in the first two months, likely due to variation in how information spreads through the community.

As shown in Fig 3A, the ABM predicted that 68% ($SD$ = 4%) of households in Perry would adopt the new fiber service. In 10,000 iterations, the minimum take rate was 52% and the maximum was 80%. According to their November 2020 dashboard report (provided by Ralls Technologies), the observed take rate was 62%. Although the model overestimated the percent of subscribers or take rate on average, this is still within the 95% confidence interval (58%–78%).

### 3.2. Predicting take rate in an unserved community

In Southeast Missouri, Bollinger County has been under the spotlight for the severity of its access gap and the need for high-speed internet [55]. In the City of Patton, there is a

**Table 3. Summary of served community scenario.**

| Attribute | Value |
|---|---|
| Number of households | 274 |
| Annual household income | *Median* = $38,000 |
| Highest household education level | *Mean* = 2.7 |
| Percent of existing users | 70% |

**Table 4. Existing and new service attributes for served community scenario.**

| Attribute | Existing Service (DSL) | New Service (Fiber) |
|---|---|---|
| Speed [Mbps] | 40 | 50 |
| Monthly cost [$/month] | $64 | $55 |
| Data cap [GB/month] | 1,024 | Unlimited |
| Reliability score | 3.2/5 (BroadbandNow) | 3.7/5 (Facebook) |

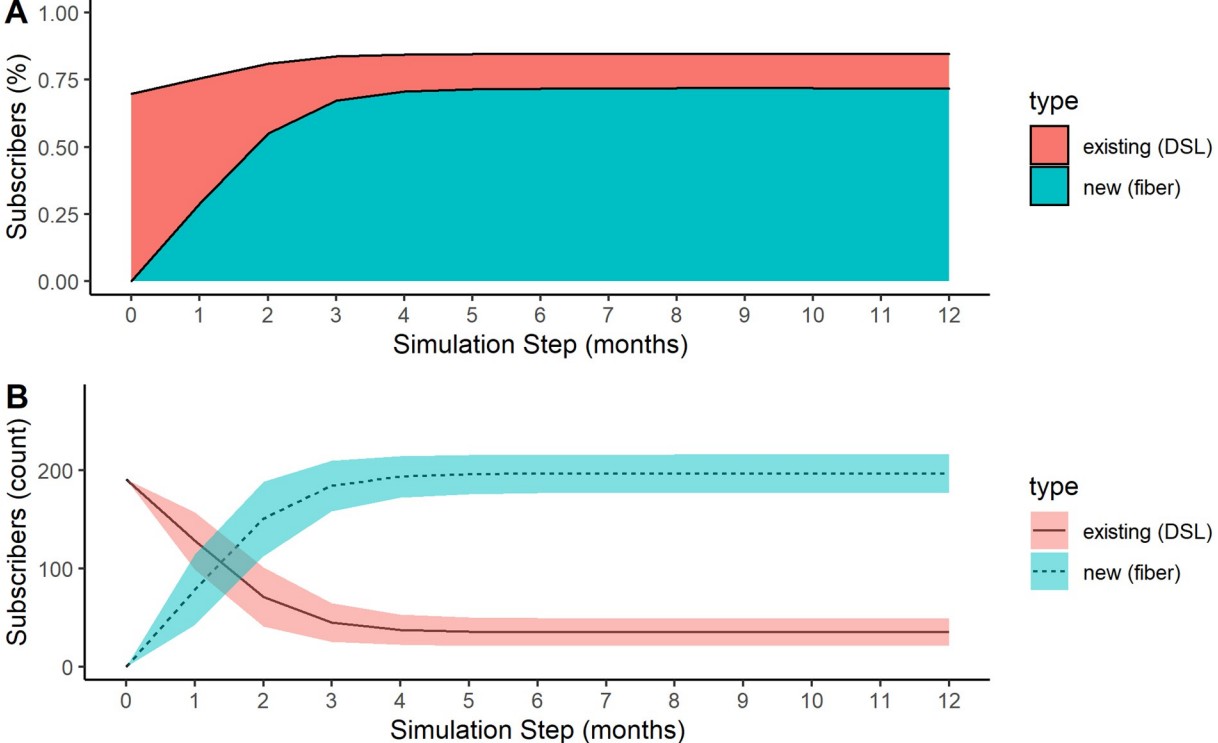

**Fig 3. Broadband subscribers over time in a served community.** Percent (A) and count (B) of subscriptions over time to existing DSL (solid, red) and a new fiber (dashed, blue) broadband service in a served community with competition. The shaded area in B represents a 95% confidence interval.

population of 462 people, with median income of $40,000 and the average resident has completed some level of high school [49]. According to the FCC Form 477, 60% of households have access through a fixed wireless or cable provider [50]. These characteristics are summarized in Table 5.

The existing fixed wireless provider is Big Rivers Communication, which has a maximum advertised speed of 7 Mbps [56]. We model 5 alternate scenarios for the new service. This area was included in the 2018 CAF-II auction, which provided $4.2 million to Wisper Internet, a fixed wireless service provider with higher speed subscription plans [57]. We compare 3 wireless scenarios with varying speeds and prices derived from Wisper Internet packages described in Table 6 [6, 58]. In addition, we compare 2 scenarios where a rural co-op provides a high-speed fiber internet connection [6].

As shown in Fig 4, the ABM predicted the highest adoption for the new fiber connections, which provided higher speeds for lower cost. The mean take rate was 78% (SD = 2%) for a

**Table 5. Summary of unserved community scenario.**

| Attribute | Value |
|---|---|
| Number of households | 462 |
| Annual household income | *Median* = $40,000 |
| Highest household education level | *Mean* = 2 |
| Percent of existing users | 60% |

**Table 6. Existing and new service attributes for unserved community scenario.**

| Attribute | Existing (Wireless) | New (Wireless) | New (Fiber) |
|---|---|---|---|
| Speed [Mbps] | 7 | 25; 50; 100 | 100; 1000 |
| Monthly cost [$/month] | $115 | $125; $130; $150 | $50; $80 |
| Data cap [GB/month] | Unlimited | Unlimited | Unlimited |
| Reliability score | 3.3/5 (BroadbandNow) | 3.2/5 (Facebook) | 4.9/5 |

1000 Mbps (1 Gbps) fiber connection and 71% (SD = 3%) for a 100 Mbps fiber connection. In addition, there was slightly higher adoption overall when fiber access was available. In contrast, we predicted much lower levels of adoption for the new wireless service. The mean take rate was 24% (SD = 3%) for a 25 Mbps wireless connection, 25% (SD = 3%) for a 50 Mbps wireless connection, and 22% (SD = 3%) for a 100 Mbps wireless connection when competing with an existing 7 Mbps wireless service. There are more differences between the new fiber and wireless service options than within those technology options.

### 3.3. Sensitivity analysis

Sensitivity analysis shows how each individual input variable influences the outcome. In the worst case, the existing service is high quality, and the new service is low quality. As a result, there is low interest in adopting the new service and estimated adoption for the new service is low. In contrast, in the best case, the new service is much higher quality than the existing

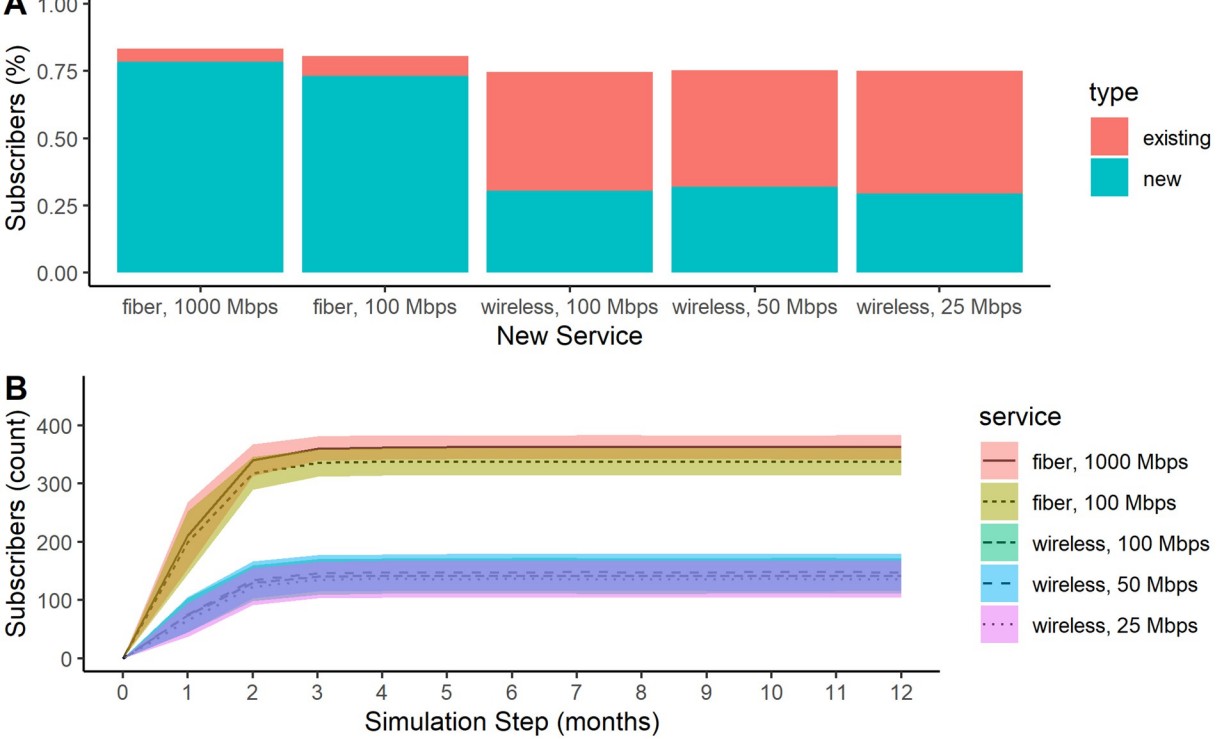

**Fig 4. Broadband subscribers over time in an unserved community.** Percent (A) and count (B) of subscriptions to existing 7 Mbps wireless vs new wireless and fiber broadband services in an unserved community. The shaded area in B represents a 95% confidence interval.

**Table 7. Sensitivity analysis of input variables when comparing the change in take rate for the worst case vs. best case.**

| Attribute | Baseline | Worst Case | | | Best Case | | | Sensitivity |
|---|---|---|---|---|---|---|---|---|
| | | Input | Take rate | % Change | Input | Take rate | % Change | |
| **Community characteristics:** | | | | | | | | |
| Existing users | 65% | 25% | 26% | -44% | 91% | 77% | 69% | **51%** |
| Total population | 500 | 10 | 0% | -100% | 1000 | 41% | -10% | **41%** |
| **New service characteristics:** | | | | | | | | |
| Price | $60 | $150 | 6% | -88% | $35 | 54% | 17% | **48%** |
| Speed | 100 | 25 | 40% | -13% | 1000 | 74% | 62% | 34% |
| Data cap | 100[a] | 30[b] | 43% | -5% | ∞[c] | 75% | 63% | 32% |
| Reliability | 3.2 | 2.8 | 32% | -28% | 4.9 | 63% | 37% | 29% |
| **Existing service characteristics:** | | | | | | | | |
| Speed | 100 | 1000 | 24% | -86% | 5 | 46% | 1% | 40% |
| Data cap | 100 | ∞ | 7% | -85% | 30 | 47% | 3% | 40% |
| Price | $60 | $150 | 26% | -44% | $35 | 64% | 40% | 38% |
| Reliability | 3.2 | 4.9 | 13% | -71% | 2.8 | 50% | 9% | 37% |
| Take rate | 46% | | | | | | | |

[a] Wireless/satellite services like T-mobile, Verizon, and NatSat offer 100 GB/month to their customers

[b] These companies also provide 1 GB/day for 4G/5G internet, which would be ~30 GB/month

[c] In the model, the unlimited data cap was represented as 3,000 GB/month so no households exceeded it

service, so the estimated adoption for the new service is high. The input values for the baseline, worst, and best cases represent observed ranges currently available in the market. In Table 7, we vary one parameter at a time from the baseline and estimate the average percent and count of subscribers for the new service over 10,000 iterations and report the percent change from the baseline. The predicted take-rate for the new service in the baseline case was 46%. The parameters with the biggest difference between the worst case to best case represent the most sensitive or influential parameters.

As highlighted in Table 7, percent of existing users, price of new service, and total population size were among the top three influential inputs. As shown in Fig 5A, the take rate increases linearly as the percent of existing users increases. For the total population size, there is much higher uncertainty in the model at lower population sizes because the stochastic elements play a larger role (see Fig 5B). However, the estimates stabilize when the population is above 300 people. This suggests that it's much more difficult to predict adoption in smaller communities.

As shown in Table 7, the quality of the existing service is more influential than the quality of the new service, except for price. Households were highly sensitive to the price of the new service. For the existing service, households were most sensitive to speed and data caps, followed by price, followed by reliability. This suggests that households were more likely to switch to the new service if (1) the new price was cheaper and (2) the service quality (i.e., speed, data cap, reliability) of the existing service was poor.

Fewer existing users switched to the new service if it was more expensive than the current service. As shown in Fig 6A, there is an inflection point at $60/month, which was the baseline value. The slopes of the lines change at $60/month, suggesting that the dynamics of the adoption pattern changed. When the new price is less than $60/month (i.e., less than the existing service), there is higher adoption of the new service (blue dotted line). When the existing service is less than $60/month (i.e., less than the new service), there is lower adoption of the new

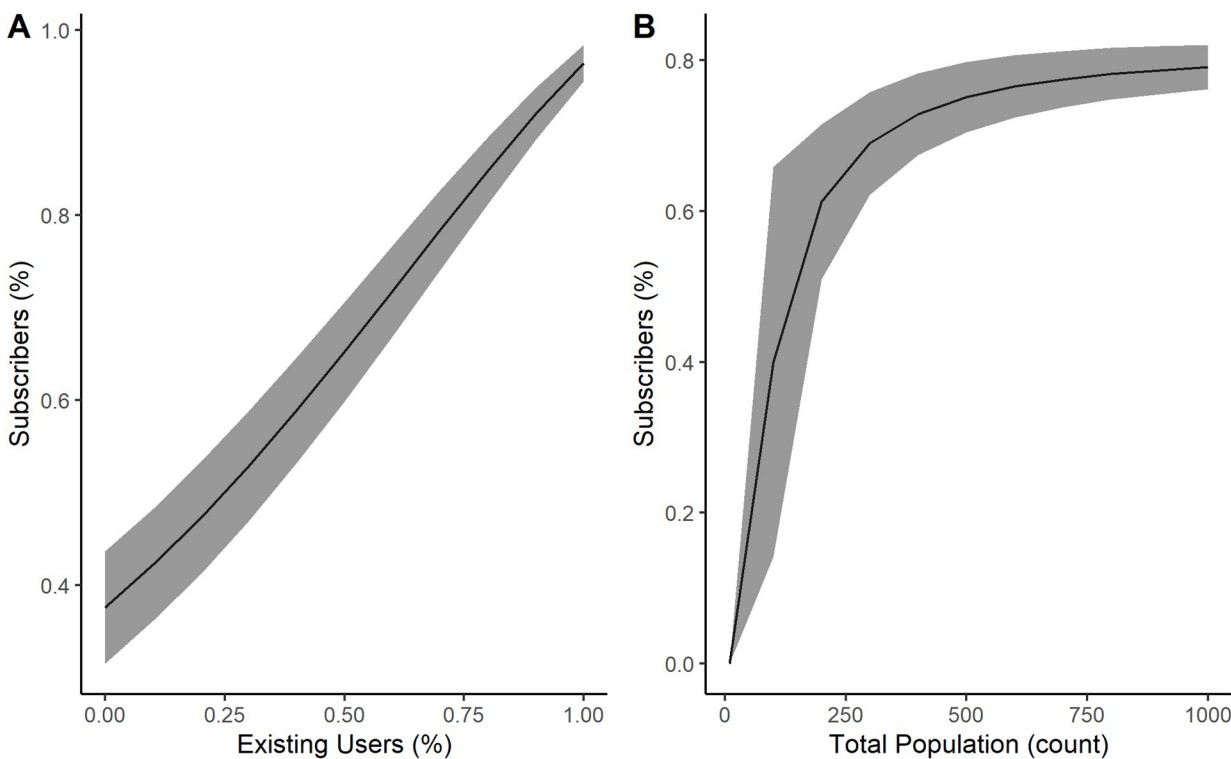

**Fig 5. Sensitivity to percent of existing users and total population.** Relationship between percent of subscribers vs. (A) percent of existing users and (B) total population size.

service (solid red line). This figure reports the percent of subscribers to the new service (not the existing service) as characteristics of the new vs. existing service changes.

As shown in Fig 6B, consumers are very sensitive to changes in speed less than 100 Mbps (i.e., the baseline speed for the new service). The slopes are very steep and then slow down at this inflection point. When the speed of the new service is less than 50 Mbps, participants prefer the existing service. However, as the speed increases (50–100 Mbps), participants start subscribing to the new service and even small improvements in speed are highly valued. After 100 Mbps, participants still prefer the new service and adoption increases beyond the existing users. However, increases after 100 Mbps show less marginal value as the slope flattens.

As shown in Fig 6C, there are two key inflection points for service reliability. When the new service has low service reliability (less than 2.5 on a scale from 1–5), consumers are less inclined to subscribe. As the service reliability of the new service approaches the existing service (i.e., 2.5 < service reliability < 3.2), consumers are increasingly inclined to subscribe to the new service. When the new service reliability is higher than the existing service reliability, most participants want the new service. In contrast, when the existing service has low reliability, most consumers subscribe to the new service. When the existing service has higher reliability than the new service, most consumers prefer the existing service.

In Fig 6D, consumers prefer higher (or unlimited) data caps. When the new service has a higher data cap than the existing service, most consumers subscribe to the new service. When the existing service has a higher data cap than the new service, most consumers stick with the existing service. However, this effect flattens out as the subscriptions approach the maximum number of interested households.

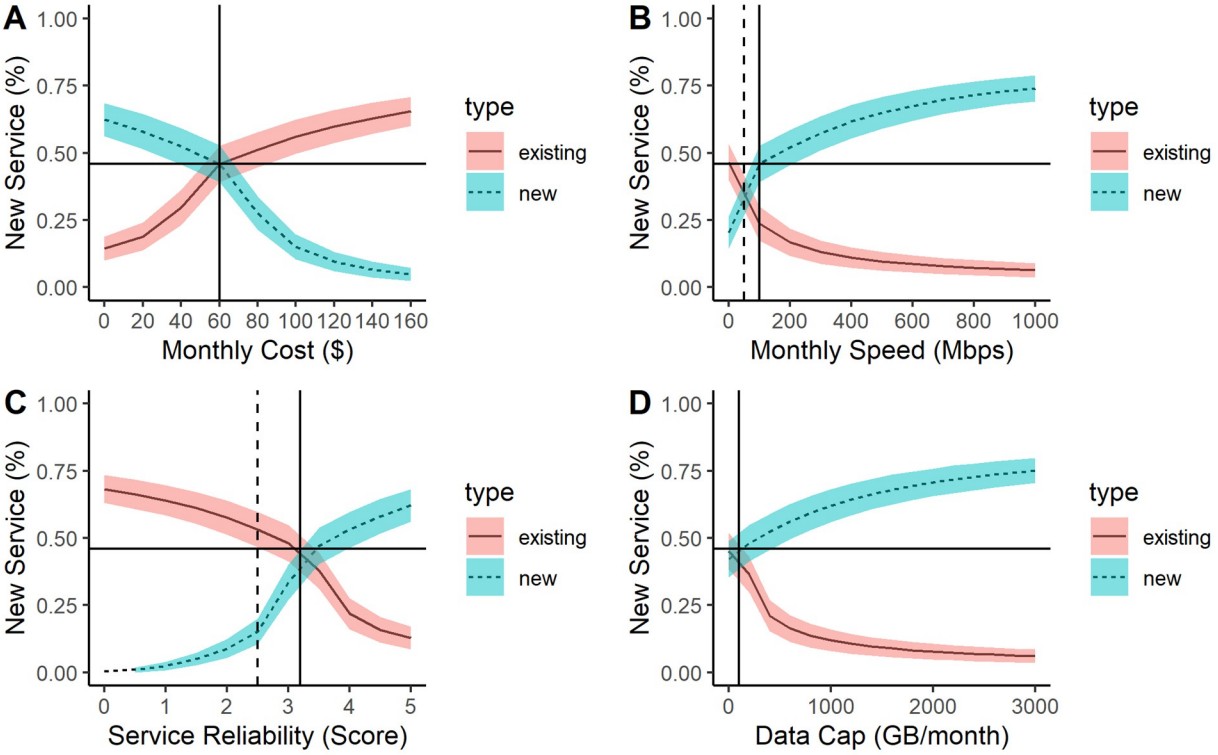

**Fig 6. Sensitivity to broadband service characteristics.** Sensitivity analysis of existing vs. new service characteristics including (A) price, (B) speed, (C) reliability, and (D) data cap on the percent of new service subscribers. The solid horizontal and vertical lines represent the baseline values. The dotted vertical lines represent additional inflection points of interest. In all cases, the percent of subscribers to the new service approaches a limit, which represents the maximum interested consumers.

## 4. Discussion

The central aim of this study revolves around crafting a simulation tool to forecast the adoption rates of high-speed broadband services within a rural locality. This simulation, employing an ABM, constructs a virtual setting comprising rural households. These households decide whether to adopt a new internet service recently introduced to their vicinity, basing their decisions on the attributes of the service and the influence exerted by their neighbors and associates. In a served community, the simulation captured the true broadband adoption rate in Perry, MO within a 95% confidence interval. This suggests that the model's outcomes are aligned with real-world observations, albeit within a singular instance. This is promising for efforts to leverage behavioral theory for configuring agent rules when modeling the phenomenon of broadband adoption. This resonates with other conceptual frameworks for technology adoption [59].

In a case study for an unserved community, we found radically different estimates of adoption depending on the quality of the broadband technology. From a policy perspective, technologies with low capital costs may be incentivized (such as wireless broadband) to maximize the number of people reached. However, our model suggests that this can lead to low adoption if this technology is associated with high costs for end-users and poor reliability. This is consistent with what was observed in many rural communities [60]. Often, technology characteristics and provider types are correlated. For example, large companies have heavily invested in fiber at the same time as they have shed rural territory [61].

Conversely, when a high-speed fiber internet service, offered at reduced monthly rates, was introduced to the agents, the adoption rates were roughly three times higher. This underscores the importance of cost-effective alternatives for consumers in fostering robust adoption rates conducive to sustainable business models. Nevertheless, even with this high-quality service, the overall adoption only marginally improved. This suggests that accessibility alone may not necessarily translate into adoption. Consumer subsidies, to address affordability, may also be critical [60, 62].

The sensitivity analysis findings suggest that rural regions boasting a substantial proportion of existing internet users are more inclined to exhibit higher adoption rates for a new service. This conclusion aligns with prior ABM literature, wherein a greater percentage of initial adopters consistently translated to a higher overall adoption rate for enhanced communication technology [46]. Furthermore, it was evident that the reduced costs associated with the new service played a pivotal role in driving broadband adoption, mirroring earlier empirical studies where pricing inversely impacted demand [25, 29]. A heightened level of utility emerges as a positive influence on adoption behavior.

There are two primary limitations to this model. First, the model predicts high uncertainty when simulating broadband adoption within sparsely populated areas. The sensitivity analysis suggests that in regions with fewer households per zip code, there are wider variations in adoption percentages. This implies that in low-density communities, a handful of individuals holding extreme views could potentially sway the entire community's decision to adopt. More research is needed to understand broadband adoption in small communities with less than 300 people. Second, another limitation lies in the simplifying assumptions that do not fully mirror the complexities of real-world market dynamics. For instance, the presence of cellular access in the area is not taken into account, and this model assumes simultaneous deployment of the new service across the entire area, rather than a phased rollout (which would introduce complications in terms of social influence). Additionally, contractual obligations may constrain consumers in their ability to switch service providers.

## 5. Conclusion

As governmental agencies actively allocate funds to ISPs for extending services into unserved and underserved regions, it becomes imperative to rigorously assess the efficacy of these policies in diminishing the digital divide [63]. The development of a simulation tool for predicting adoption rates in broadband expansion projects can serve a dual purpose. It not only assists ISPs in discerning the optimal subscription plans to offer but also empowers policymakers in the strategic allocation of resources. The present study underscores the value of simulation models in facilitating the prediction of consumer behavior for rural broadband and bridging the digital divide. This can support rural communities taking an active role in procuring broadband for their region [64].

Future research efforts should focus on refining simulations prior to utilization for bolstering evidence-based policymaking [43]. Particular attention should be devoted to obtaining data related to the spatial availability of broadband service, consumer preferences, and customer service metrics. Emerging data sources, such as the broadband fabric, hold the potential to significantly elevate the utility of ABMs by providing more granular geospatial information [65]. Incorporating surveys to initialize the ABM could be invaluable [66, 67]. For instance, discrete choice experiments have proven effective in ascertaining consumer preferences regarding e-groceries in urban settings [68], solar PV adoption in New Zealand [69], and the availability of wood in Swiss markets [70]. In addition, the inclusion of customer service attributes such as contractual terms, one-time costs, quality of service, and latency would be

valuable. Ultimately, behavioral simulations like ABMs hold the potential for synergistic integration with other modeling initiatives, such as economic development models, to inform policymaking and the formulation of market-based incentives.

## Acknowledgments

We would like to thank Melinda Stormes, Andrew Blanton, Mark Keeling and Carmen Hartwell of Gascosage Electric Cooperative for lending their precious time and advice about rural broadband adoption. We also acknowledge support from Lynn Hodges of Ralls Technologies for sharing data. This work is derived from Ankit Agarwal's master's thesis.

## Author Contributions

**Conceptualization:** Ankit Agarwal, Casey Canfield.

**Data curation:** Ankit Agarwal.

**Formal analysis:** Ankit Agarwal.

**Investigation:** Ankit Agarwal, Casey Canfield.

**Methodology:** Ankit Agarwal, Casey Canfield.

**Project administration:** Casey Canfield.

**Software:** Ankit Agarwal.

**Supervision:** Casey Canfield.

**Validation:** Casey Canfield.

**Visualization:** Ankit Agarwal.

**Writing – original draft:** Ankit Agarwal.

**Writing – review & editing:** Casey Canfield.

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
