## [Decision Letter · Decision Letter 0]

6 Feb 2024

PONE-D-23-44061Using a Theory-Driven Agent-Based Model to Explain Rural Broadband Adoption DynamicsPLOS ONE

Dear Dr. Canfield,

Thank you for submitting your manuscript to PLOS ONE. After careful consideration, we feel that it has merit but does not fully meet PLOS ONE’s publication criteria as it currently stands. Therefore, we invite you to submit a revised version of the manuscript that addresses the points raised during the review process.

We look forward to receiving your revised manuscript.

Kind regards,

Muhammad Nasir Khan, PhD

Academic Editor

PLOS ONE

Reviewers' comments:

Reviewer's Responses to Questions

**Comments to the Author**

1. Is the manuscript technically sound, and do the data support the conclusions?

Reviewer #1: Yes

Reviewer #2: Yes

2. Has the statistical analysis been performed appropriately and rigorously? 

Reviewer #1: Yes

Reviewer #2: Yes

3. Have the authors made all data underlying the findings in their manuscript fully available?

Reviewer #1: Yes

Reviewer #2: Yes

4. Is the manuscript presented in an intelligible fashion and written in standard English?

Reviewer #1: Yes

Reviewer #2: Yes

5. Review Comments to the Author

Reviewer #1: “Using a Theory-Driven Agent-Based Model to Explain Rural Broadband Adoption Dynamics”

First and foremost, I suggest that you rephrase your full title to “Analysis of Rural Broadband Adoption Dynamics: A Theory-Driven Agent-Based Model.”

Referring to the introduction section, you may use questions to state your research question, or to clarify an issue that requires future research. Normally, however, you should avoid using questions in your formal or academic writing.

Do not use “i)” within paragraphs. Instead, use discourse markers. Similarly, the discussion section includes “a)” etc. in-text. Please note that these are used as bullets. Use discourse markers like “one, secondly, moreover” only.

In the introduction section, there are many statements that do not have a reference. Make sure that you reference every statement properly. For example, the paragraph where you have talked about the pandemic has zero in-text citations.

As per PLOS guidelines, the introduction section must include a brief review of the key literature. I found that in the start of the “Modelling Broadband Adoption” section.

It is imperative to include the GitHub repository URL in references instead of in the main text. Simply include the author and year of publication as in-text citation. The URL goes in the corresponding reference list entry.

Kindly rephrase figure 1 title as per the PLOS guidelines. Include the description as part of the legend (instead of the title). Same case in figures 3, 4, and 6.

Lastly, for clarity purpose, separate the discussion and conclusion sections. Alternatively, make sub-sections in the mixed section.

All in all, this is a very good paper and has considerable potential to get published. Good luck!

Reviewer #2: 1. Rewrite the abstract to reflect the work.

2. The introduction is lack of research gap and questions.

3. I the model is design by authors?

4. What re the contributions?

5. What type of software or technology are used for the implementation of model?

6. The author should compare the results with literature.

6. PLOS authors have the option to publish the peer review history of their article (what does this mean?). If published, this will include your full peer review and any attached files.

Reviewer #1: **Yes: **Bilal Aftab

Reviewer #2: No

---

## [Author Response · Author response to Decision Letter 0]

27 Mar 2024

Dr. Khan,

Thank you for the opportunity to revise our manuscript. The feedback from the reviewers has greatly improved this work. In particular, we have substantially revised the introduction and discussion to better frame and contextualize our findings. In addition, we have reviewed the manuscript for grammatical errors and identified opportunities to clarify the language.

Reviewer 1

Comment R1.1. First and foremost, I suggest that you rephrase your full title to “Analysis of Rural Broadband Adoption Dynamics: A Theory-Driven Agent-Based Model.”

Response R1.1. Thank you for the suggestion - we have revised the title.

Comment R1.2. Referring to the introduction section, you may use questions to state your research question, or to clarify an issue that requires future research. Normally, however, you should avoid using questions in your formal or academic writing.

Response R1.2. We have removed the rhetorical questions from the introduction. Instead, we now pose research questions.

Comment R1.3. Do not use “i)” within paragraphs. Instead, use discourse markers. Similarly, the discussion section includes “a)” etc. in-text. Please note that these are used as bullets. Use discourse markers like “one, secondly, moreover” only.

Response R1.3. We have revised the manuscript to use discourse markers. 

Comment R1.4. In the introduction section, there are many statements that do not have a reference. Make sure that you reference every statement properly. For example, the paragraph where you have talked about the pandemic has zero in-text citations.

Response R1.4. We have added 13 citations to the introduction to support our statements and revised the text to improve clarity.

Comment R1.5. As per PLOS guidelines, the introduction section must include a brief review of the key literature. I found that in the start of the “Modelling Broadband Adoption” section.

Response R1.5. We have revised the structure of the manuscript to make the literature review a subsection in the Introduction, rather than a separate section.

Comment R1.6. It is imperative to include the GitHub repository URL in references instead of in the main text. Simply include the author and year of publication as in-text citation. The URL goes in the corresponding reference list entry.

Response R1.6. We have moved the GitHub URL to a citation.

Comment R1.7. Kindly rephrase figure 1 title as per the PLOS guidelines. Include the description as part of the legend (instead of the title). Same case in figures 3, 4, and 6.

Response R1.7. We have revised the captions to separate the title and legend.

Comment R1.8. Lastly, for clarity purpose, separate the discussion and conclusion sections. Alternatively, make sub-sections in the mixed section.

Response R1.8. We have separated the Discussion and Conclusion sections.

Comment R1.9. All in all, this is a very good paper and has considerable potential to get published. Good luck!

Response R1.9. Thank you for your thoughtful comments!

Reviewer 2

Comment R2.1. Rewrite the abstract to reflect the work.

Response R2.1. Thank you for your thoughtful comments! We have revised the abstract to better reflect the results and improve the clarity of the language.

Comment R2.2. The introduction is lack of research gap and questions.

Response R2.2. We have added a paragraph to the introduction to highlight the research questions and research gap. 

“This research aims to address two primary research questions:

1. Can a theory-based simulation capture broadband adoption dynamics in served and unserved communities?

2. How do attributes of the broadband technology and population influence adoption dynamics?

The primary contribution of this work is in demonstrating an approach for scaling modeling efforts for broadband adoption in data poor contexts. In the broadband context specifically, this requires modeling market competition, social network effects, digital literacy, and affordability.”

Comment R2.3. I the model is design by authors?

Response R2.3. Yes, the model is designed and implemented by the authors.

Comment R2.4. What re the contributions?

Response R2.4. The primary contribution of this work is in demonstrating an approach for scaling modeling efforts for broadband adoption in data poor contexts. We have added this context to the introduction.

Comment R2.5. What type of software or technology are used for the implementation of model?

Response R2.5. The model is implemented in NetLogo. This is reported in the Model overview section.

Comment R2.6. The author should compare the results with literature.

Response R2.6. We have revised the discussion section by adding additional citations to contextualize our results.

---

## [Editor Report · Decision Letter 1]

28 Mar 2024

Analysis of Rural Broadband Adoption Dynamics: A Theory-Driven Agent-Based Model

PONE-D-23-44061R1

Dear Dr. Canfield,

We’re pleased to inform you that your manuscript has been judged scientifically suitable for publication and will be formally accepted for publication once it meets all outstanding technical requirements.

Kind regards,

Muhammad Nasir Khan, PhD

Academic Editor

PLOS ONE
---

## [Editor Report · Acceptance letter]

13 May 2024

PONE-D-23-44061R1 

PLOS ONE

Dear Dr. Canfield, 

I'm pleased to inform you that your manuscript has been deemed suitable for publication in PLOS ONE. Congratulations! Your manuscript is now being handed over to our production team.

Kind regards, 

on behalf of

Dr. Muhammad Nasir Khan 

Academic Editor

PLOS ONE